# MULTIMEDIA-AGENT: A MULTIMODAL AGENT FOR MULTIMEDIA CONTENT GENERATION

## ABSTRACT

With the advancement of AIGC (AI-generated content) technologies, an increasing number of generative models are revolutionizing fields such as video editing, music generation, and even film production. However, due to the limitations of current AIGC models, most models can only serve as individual components within specific application scenarios and are not capable of completing tasks end-to-end in real-world applications. In real-world applications, editing experts often work with a wide variety of images and video inputs, producing multimodal outputs—a video typically includes audio, text, and other elements. This level of integration across multiple modalities is something current models are unable to achieve effectively. However, the rise of agent-based systems has made it possible to use AI tools to tackle complex content generation tasks. To deal with the complex scenarios, in this paper, we propose a multimedia content generation agent system designed to automate complex content creation. Our agent system includes a data generation pipeline, a tool library for content creation, and a set of metrics for evaluating preference alignment. Notably, we introduce the skill acquisition theory to model the training data curation and agent training. We designed a two-stage correlation strategy for plan optimization, including self-correlation and model preference correlation. Additionally, we utilized the generated plans to train the MultiMedia-Agent via a three stage approach including base/success plan finetune and preference optimization. The comparison results demonstrate that the our approaches are effective and the MultiMedia-Agent can generate better multimedia content compared to GPT4o.

## 1 INTRODUCTION

AI-generated content, such as images, videos, audio, etc., has gradually been applied to various aspects of everyday life (Liu et al., 2024a; Esser et al., 2024). However, the needs of the real world are complex and diverse. Taking the field of video generation as an example, a user's input may not only be text or a single image but could also include materials like images, music, etc, and the model needs to integrate these materials to generate an appropriate video. Moreover, on the output side of the model, the video that the user requires may not only consist of video frames but also include suitable background music, voiceovers, subtitles, and so on. Clearly, a single generative model at present cannot accomplish this task. One possible solution to handle such complex situations is to use an agent system to understand user needs while integrating different downstream tools to process complex inputs and outputs (Wang et al., 2024c;a). Some existing agent systems (Shen et al., 2024) can call upon multiple tools to handle content generation, but they are not specifically designed for content generation, nor do they take into account the diverse needs of real life.

Therefore, in this paper, we will explore whether multimodal agent can learn such complex workflows of multi-media content creation in a manner similar to humans. Specifically, we will investigate whether the multimodal agent can progressively acquire complex skills from scratch, following the stages of **skill**

**acquisition theory** (DeKeyser, 2020), which mirrors the way of how humans learn skills step by step. Precisely, skill acquisition theory consists of three stages:

1. **Cognitive Stage**: In this stage, beginners need to learn the fundamental operations and knowledge, attempting to understand the basic concepts related to the new skill.

2. **Associative Stage**: At this stage, learners begin to engage in conscious, targeted practice, refining their skills through repeated operations.

3. **Autonomous Stage**: To reach this stage, learners require continuous feedback and improvement, including both self-correction and external supervision.

We first developed a multi-media content playground based on real-world scenarios, incorporating a feedback mechanism. This playground includes 18 common multimodal generation scenarios tailored to real-world needs and features a content creation tool library that supports the editing, generation and retrieval of images, videos, audio, speech, and text. Then, to provide data for training the multimodal agent, we constructed different levels of plans from the perspective of the three stages of skill acquisition theory. First, we used GPTo as a teacher to generate a base plan for each question within each task. Of course, this base plan may not always execute successfully, so further optimization is needed. In the second step, we had GPT4o reflect on and perform self-correction on the base plan generated in the first stage, thereby improving the plan's quality. In the final step, we introduced external preference models to evaluate the plan from the second step, and then allowed GPT4o to optimize it further. In this way, we obtained three different levels of plans to train the multimodal agent.

During the training of the multimodal agent, we followed the three stages of Skill Acquisition Theory. First, in the Model Cognitive Stage, we fine-tuned the agent using all the generated plans. This allowed the agent to quickly grasp the purposes of the tools, their output formats, and basic operational principles—similar to how a human beginner learns foundational knowledge and basic concepts from a large amount of data. In the Model Associative Stage, we fine-tuned the agent using only successfully executed plans. At this stage, the agent learns more advanced logic, such as the composition of workflows and the relationships between tools, building upon its foundational understanding of tool usage. Finally, in the Model Autonomous Stage, we performed post-training using paired preference data constructed based on the model's preferences. This stage enables the multimodal agent not only to complete tasks but also to perceive and apply emotional or aesthetic needs, such as human preferences, to tool execution.

It is worth noting that the MultiMedia-Agent differs significantly from previous models and methods. We present the differences between MultiMedia-Agent, other tool-agents, and content creation agent systems in the Table 1. Most of the methods listed in the table support multimodal content generation. For multimodal content understanding, HuggingGPT (Shen et al., 2024) can indirectly handle multimodal understanding by invoking APIs, NExT-GPT (Wu et al., 2023c) and MLLM-Tool can directly understand multimodal data. AutoDirector (Ni et al., 2024) can only process text input and cannot generate content based on user-provided materials. While our model is capable of accepting multi-modal and multiple inputs. In terms of planning ability, ToolLLM (Qin et al., 2023) and HuggingGPT can plan the use of multiple tools based on user instructions, whereas NExT-GPT and ModaVerse (Wang et al., 2024c) can only output a single piece of content in one forward pass. Multimodal Interaction refers to whether the model can generate and integrate multiple types of modalities based on user needs. Currently, only works like AutoDirector have such capabilities but only limited to video scenario. Our MultiMedia-Agent can handle the plan generation for multiple scenarios based on the user's query. As for content generation scenarios, none of the existing models perform preference alignment on the generated content, whereas MultiMedia-Agent incorporates human-preference-based evaluation models to handle this aspect.

In summary, our contributions are as follows:

| | Multimodal Generation | Multimodal Understanding | Planning Ability | Multimodal Interaction | Preference Alignment |
|---|---|---|---|---|---|
| ToolLLM | ✗ | ✗ | ✓ | ✗ | ✗ |
| HuggingGPT | ✓ | ✗ | ✓ | ✗ | ✗ |
| NExT-GPT | ✓ | ✓ | ✗ | ✗ | ✗ |
| ModaVerse | ✓ | ✓ | ✗ | ✗ | ✗ |
| AutoDirector | ✓ | ✗ | ✓ | ✓ | ✗ |
| MultiMedia-Agent | ✓ | ✓ | ✓ | ✓ | ✓ |

Table 1: A comparison of our MultiMedia-Agent with notable tool agents or content creation agents.

1. We build a plan generation system for multi-media content generation, including a data generation pipeline, tool library and evaluation metrics.

2. We design a two-stage correlation of plan curation specifically for multi-media content generation according to skill acquisition theory. Utilizing self-correlation and preference model correlation strategies for plan optimization.

3. We propose a three-stage training pipeline for multimedia content generation agent based on skill acquisition theory. This pipeline enables the agent to effectively learn the generation of complex plans from scratch. The agent demonstrated exceptional performance across various tasks.

## 2 RELATED WORK

### 2.1 TOOL AGENT

With the rise of LLM agents (Mei et al., 2024; Liu et al., 2023; 2024b; Zhao et al., 2024), enabling agents to call external APIs to solve user problems has become a crucial research topic. Toolformer (Schick et al., 2024) pioneered the exploration of connecting large language models (LLMs) with external tools. HuggingGPT (Shen et al., 2024) leveraged an agent to call HuggingFace's API, allowing it to solve a wide range of complex problems. Subsequent research has extended this integration to fields like healthcare support (Ma et al., 2023b), code synthesis (Wang et al., 2024b), and web searching (Ma et al., 2023a). ToolLLM (Qin et al., 2023) focused on executing complex tasks in real-world scenarios. GPT4Tools (Yang et al., 2024) and Visual ChatGPT (Wu et al., 2023a) integrated visual foundation models after decomposing tasks into manageable components. For multimodal tool agents, MLLM-Tool (Wang et al., 2024a) employed multimodal large models as agents to call Hugging Face APIs. Similarly, ModaVerse (Wang et al., 2024c) used multimodal large models for any-to-any generation. In our MultiMedia-Agent, we focus primarily on the planning and alignment capabilities of multimodal agents, aiming to enhance content generation quality.

### 2.2 ANY-TO-ANY GENERATION

The earliest any-to-any model was CoDi (Tang et al., 2024), followed by NextGPT and EMU (Sun et al., 2023), which introduced further improvements in data and model design. EMU2 (Sun et al., 2024) introduced a unified autoregressive objective to predict the next multimodal element, either by regressing visual embeddings or classifying textual tokens. CM3Leon (Yu et al., 2023) and Chameleon (Team, 2024) used mixed image and text data to train token-based autoregressive models. More recently, TransFusion (Zhou et al., 2024) and Show-o (Xie et al., 2024) combined large language models with diffusion models to enhance performance.

However, any-to-any models are typically limited to generating a single modality without considering the relationships and connections between modalities. This is precisely the area that our MultiMedia-Agent focuses on, emphasizing the interplay between different modalities for richer content generation.

| Audio/Video to Audio | Audio/Video to Text | Audio/Video to Video |
|---|---|---|
| Image/Audio to Text | Image/Audio to Video | Image/Video to Audio |
| Image/Video to Text | Image/Video to Video | Multiple Audios to Image |
| Multiple Audios to Text | Multiple Audios to Video | Multiple Images to Audio |
| Multiple Images to Text | Multiple Images to Video | Multiple Videos to Audio |
| Multiple Videos to Image | Multiple Videos to Text | Multiple Videos to Video |

Table 2: 18 real world task types.

## 3 DATA CURATION

In this section, we will systematically analyze the procedures we took to construct our dataset. To generate complex multi-modal media content, we first built a multimodal tool library, from which the agent can select appropriate tools to form the plan. Next, we construct differentiated plans based on different types of requests and feedbacks. Finally, we designed a series of metrics to evaluate and provide model preference feedback of the content generated by these plans, thereby enabling the assessment and ranking of the plans.

### 3.1 MULTI-MEDIA TASKS

Due to the lack of datasets that can model real-world multimodal demand-solution scenarios, as shown in Table 2, we first constructed scenarios based on various real-world needs. For example, *A user might want to automatically convert a series of photos into a video slideshow, possibly for a wedding or event photo montage.* This can be summarized as a multi-images-to-video task. Another scenario could be: *A person has taken some photos and wants to use them along with selected background music to generate a video, like creating a travel memory video.* This can be categorized as an image-audio-to-video task. In this case, we designed 18 types of tasks, involving modalities such as image, video, audio, speech, and text. Next, we constructed user queries for each task and collected corresponding multimedia data. Specifically, we first gathered publicly available multimedia data from the web, then used GPT-4o to generate user queries under different circumstances by combining the multimedia data with task type information. As a result, we built a diverse dataset of multimedia tasks, with corresponding user queries and multimedia data for each task.

### 3.2 TOOL LIBRARY

Considering the complex relationships and connections between different modalities, we built this tool library from three main perspectives: Multimodal Understanding Tools, Multimodal Generation/editing Tools, and Auxiliary Tools. We present the whole tool library in Appendix A.1.

**Multi-modal Understanding Tools.** A good agent system should first perceive the environment before taking action. Therefore, we designed understanding models for each modality, enabling the agent system to perceive information from different types of modal data, leading to better plan curation. Specifically, we introduced five any-to-text models, corresponding to the five modalities: image, video, speech, audio, and text.

**Generative/editing Tools.** At the same time, our agent system requires the capability to generate and edit data across different modalities. Therefore, we introduced a suite of generation and editing tools, encompassing the creation and modification of images, video, audio, and speech. Additionally, we incorporated several non-deep learning tools, such as video transition effects and audio effects, to provide comprehensive multimedia editing capabilities.

**Auxiliary Tools.** In addition, we introduced Auxiliary tools, which include essential multimedia data processing utilities that cannot be overlooked, such as tools for video-to-video concatenation, video-to-audio synchronization, video retrieval tools and other basic operations.

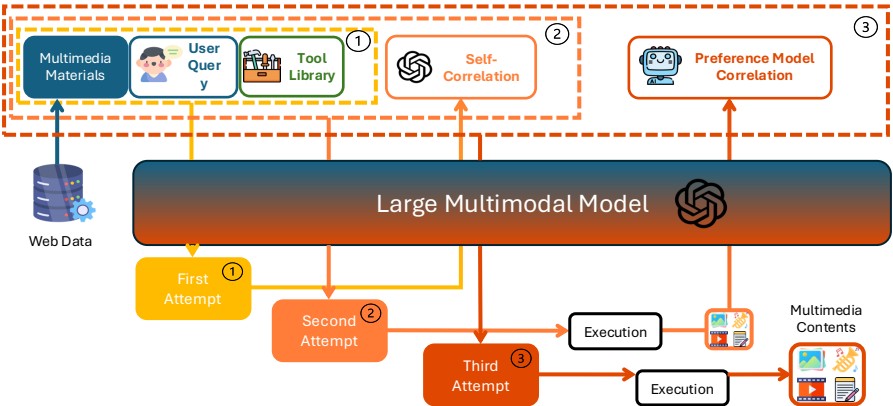

Figure 1: Two-stage correlation of plan curation for content creation.

We organized the information for each tool into JSON file. The keys in the prompt consist of the following part: **Tool name and execution model name:** We first defined the tool names and their corresponding models. When designing the tool names, we considered that our agent involves multiple modalities and various input-output models, which can easily lead to incorrect file formats being generated during the planning stage. To address this, we fixed the file formats for the four modalities as follows: Image: *.png*; Video: *.mp4*; Audio & Speech: *.mp3*; Text: *.txt*. We also included both the input and output formats in the tool name, for example, *text_txt_to_video_mp4*, to ensure more stable plan curation. Additionally, the JSON file defines the model names associated with each tool, which are used to index the models during execution. **Required parameters:** Due to the complexity of our tasks and data, for each required parameter, we provide a detailed description of its purpose. For example, in the object removal tool, where the input parameters include a text description and the input image name, the required input parameters are defined as: {*"text", "description of the object to be removed"*} and {*"image", "image file from which object needs to be removed"*}. This approach helps ensure that the model can output the tool information more accurately. **Tool Description:** Including the functionality and description of the tool is essential to better prompt the agent model.

### 3.3 HIERARCHICAL PLAN CURATION

Once we have constructed the user queries, corresponding multimedia data, and the tool library, we utilized GPT-4o to generate the plans. The plans are organized into a list of dictionaries, where each dictionary includes the information of the tool.

To enable the multimodal agent to better learn complex skills, we need to incorporate feedback into the model's training data. Unlike conventional tool agents, content-generation-oriented tool agents must not only consider the execution success rate of the plan but also ensure that the generated multimedia content meets human needs and aesthetic standards. In other words, the agent we are training must not only possess the ability to complete complex tasks using tools but also be responsible for the outcomes produced by these tools, ensuring that the results align with human needs and preferences. To address this, we designed a two-stage correlation approach for tool plan curation. After generating the base plan, we first employ GPT4o to perform self-correction, identifying issues within the plan and optimizing it to obtain a self-corrected plan. Next, we execute the self-corrected plan to produce the multimedia result. We then employ a series of model-based preference evaluation metrics to assess the quality of the multimedia result. Using these metrics, the LMM further refines the plan to optimize it, ultimately yielding the final plan.

### 3.3.1 TWO-STAGE CORRELATION OF PLAN CURATION

**Stage 1: Self-correlation**

After inputting the user query, materials, and tool information into GPT-4o to generate the base plan, we further prompt GPT-4o to evaluate the quality of the current plan based on all available information. Our evaluation criteria focus on two main aspects: **Plan execution success rate:** We prompt GPT-4o to assess whether the plan can be successfully executed, and if not, make necessary modifications. **Inclusion of user-requirement-aligned, common-sense optimizations:** We assess whether the plan includes additional optimization tools that meet the user's needs and adhere to common sense, such as adding background music to a video or incorporating voiceovers in audio generation.

This ensures the plan is both executable and aligned with user expectations.

**Stage 2: Model Preference Correlation**

To further evaluate the generated plans, in Stage 2 we assess the results of content generation plans using model-based preference feedback metrics for the four output modalities: image, video, audio, and text. Our evaluation focuses on three primary aspects: whether the generated multimedia content meets human needs, conveys emotional expression, and aligns across modalities.

- **Text output metrics.** We use GPT-4o to evaluate the alignment between the input and output content.

- **Image output metrics.** GPT-4o assesses whether the images meet human needs and convey emotions, while Pick Score Kirstain et al. (2023) is used to evaluate aesthetics.

- **Audio output metrics.** We apply speech-to-text and audio-to-text models to convert the audio into text, and GPT-4o evaluates the fulfillment of human needs and emotional expression based on user requirements and input content.

- **Video output metrics.** Similar to image outputs, GPT-4o evaluates whether the video meets human needs and conveys emotions, while Dover Score (Wu et al., 2023b) is used for aesthetic and quality evaluation. If the video includes embedded audio, we apply the same evaluation methods used for audio outputs. Additionally, we introduce a audio-video alignment metric, where GPT-4 scores the alignment between the transcribed audio text and the video content.

By integrating these metrics, we provide a comprehensive evaluation of any type of plan execution output, reflecting the overall quality of the plan. We use the optimized plans from Stage 1 to generate multimedia content and then apply the above metrics for evaluation. The evaluation results are fed back to GPT-4o, which, based on the feedback and previous information, generates a new plan. We show a generated plan in the Appendix A.2.

### 3.4 DATA STATISTICS

In this section, we primarily present the statistics of the dataset we constructed, including success rate, average steps, and average metrics. For each task type, we generated 70 user requests under different conditions. For each individual user request, we constructed three plans. We first calculated the number of steps in each plan for each task type, as shown in Table 3. Here, "T" represents "Text," "A" represents "Audio," "V" represents "Video," "I" represents "Image," and "M" represents "Multi-input." We can observe that for more complex tasks like video generation, the plans tend to include more steps to complete the task, whereas for text generation, the model requires fewer steps to accomplish the task. Additionally, as the plans are optimized, the number of steps required to complete the tasks increases, indicating that Self-correlation and Model Preference Correlation introduced more tool usage in the plan generation process.

We further illustrate the success rates of the generated plans in the Table 4. When combined with the number of steps in each plan, it becomes evident that as the number of steps in a plan increases, the success rate of the plan decreases.

Table 3: Average steps for different tasks.

|  | AV-A | AV-T | AV-V | IA-T | IA-V | IV-A | IV-T | IV-V | MA-I |
|---|---|---|---|---|---|---|---|---|---|
| Plan 1 | 5.8 | 2.9 | 4.1 | 3.0 | 4.8 | 4.3 | 3.0 | 6.3 | 5.1 |
| Plan 2 | 6.1 | 3.1 | 5.4 | 3.0 | 5.6 | 8.4 | 3.0 | 6.6 | 5.2 |
| Plan 3 | 6.2 | 3.1 | 5.6 | 3.0 | 6.2 | 9.2 | 3.1 | 7.8 | 6.3 |

|  | MA-T | MA-V | MI-A | MI-T | MI-V | MV-A | MV-I | MV-T | MV-V |
|---|---|---|---|---|---|---|---|---|---|
| Plan 1 | 4.0 | 8.1 | 4.2 | 4.1 | 8.5 | 7.6 | 12.0 | 4.1 | 6.0 |
| Plan 2 | 4.1 | 9.4 | 5.5 | 4.1 | 8.8 | 8.0 | 12.2 | 4.1 | 6.4 |
| Plan 3 | 4.4 | 11.8 | 8.2 | 4.5 | 10.6 | 9.2 | 12.8 | 4.1 | 7.4 |

Table 4: Success rate (%) for different tasks.

|  | AV-A | AV-T | AV-V | IA-T | IA-V | IV-A | IV-T | IV-V | MA-I |
|---|---|---|---|---|---|---|---|---|---|
| Plan 1 | 100 | 100 | 100 | 100 | 100 | 100 | 100 | 100 | 91.42 |
| Plan 2 | 100 | 100 | 100 | 100 | 90.00 | 98.57 | 100 | 88.57 | 97.14 |
| Plan 3 | 90.00 | 98.60 | 10.00 | 100 | 74.29 | 91.42 | 100 | 12.86 | 92.86 |

|  | MA-T | MA-V | MI-A | MI-T | MI-V | MV-A | MV-I | MV-T | MV-V |
|---|---|---|---|---|---|---|---|---|---|
| Plan 1 | 100 | 47.14 | 100 | 100 | 74.29 | 91.42 | 90.00 | 100 | 90.00 |
| Plan 2 | 100 | 42.86 | 100 | 100 | 47.14 | 91.42 | 90.00 | 100 | 91.43 |
| Plan 3 | 85.71 | 24.28 | 67.14 | 85.71 | 27.14 | 95.71 | 71.42 | 98.60 | 80.00 |

## 4 MULTIMEDIA-AGENT

### 4.1 AGENT SKILL ACQUISITION

We further used our data to train an multimodal agent. To better encode the tool information and the behaviors from the plan into the multimodal model, we designed a three-stage training approach based on skill acquisition theory.

1. **Model Cognitive Stage.** At this stage, the agent primarily focuses on learning the basic usage of tools and understanding the input-output JSON formats. We trained the model using all available data.

2. **Model Associative Stage.** At this stage, we trained the model using only successful plans, the agent begins to learn established action trajectories from successfully executed plans to ensure smooth execution and accurate output of future plans.

3. **Model Autonomous Stage.** At this stage, the agent not only needs to develop the ability to synthesize complex plans but also must ensure that the generated content aligns with human aesthetics and preferences. So, we categorized the plans into winning and losing plans based on the metric model's scores. Then, we applied DPO (Direct Preference Optimization) to align the model with these preferences.

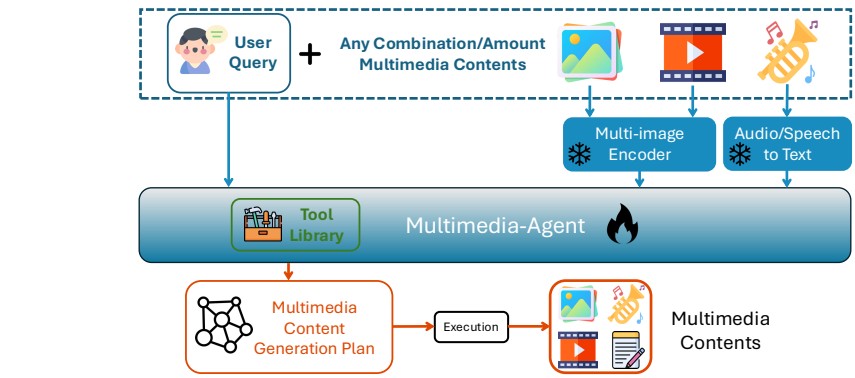

Figure 2: The detailed structure of MultiMedia-Agent.

## 4.2 EXPERIMENT SETTINGS

We use Minicpm-v2 (Yao et al., 2024) as our backbone to train out MultiMedia-Agent. The agent structure is shown in Figure 2. Specifically, when processing videos, we extract 3 evenly spaced frames to represent the entire video. Since the number of input videos and images is not fixed, we concatenate all the images into a single large image before feeding it into the model. For audio and speech, we first use audio-to-text or speech-to-text models to convert the input into text, which is then passed into the LLM for further processing. The training details are attached in the Appendix. For model validation, we generated an additional 10 user queries for each task and used GPT-4o as a comparison method. The metrics we selected for evaluation included not only success rate but also model preference feedback.

## 4.3 ANALYSIS OF THE RESULT

### 4.3.1 SKILL ACQUISITION THEORY CAN BENEFIT TOOL AGENT TRAINING

We first present a comparison between our MultiMedia-Agent and GPT-4o in terms of success rate. As observed, the agent trained through the Model Associative Stage shows a significant improvement in success rate. However, upon completion of the Model Autonomous Stage, we noticed a decline in success rate. This may be due to the tendency of the model to generate longer plans after the Model Autonomous Stage, and given the model's limited capacity, errors are more likely to occur when generating extended text. This issue is also evident from the comparisons shown in Table 3 and Table 4. This highlights that ensuring the agent outputs a stable plan format is a key challenge for tool agents when dealing with complex scenarios.

We further analyzed the model preference feedback results for content generated by MultiMedia-Agent at different stages compared to GPT-4o. All reported metric results are the average model feedback scores for successfully executed plans.

Due to space limitations, we only present the results for text generation and video generation here. Other reults are presented in Appendix A.4. MultiMedia-Agent-1/2/3 correspond to the agents after each of the three training stages, respectively. As shown in the table, after Stage 1 (Model Cognitive Stage), the MultiMedia-Agent produces fairly average results. Following the second stage of training (Model Associative Stage), the scores dropped, which may be due to the fact that successful plans tend to have fewer steps, leading to weaker alignment. Moreover, after Stage 3 (Model Autonomous Stage), MultiMedia-Agent showed significant improvements across various metrics. This demonstrates that the rewards from the preference model can effectively optimize the tool agent's plan generation. Our three-stage training enables the model to effectively learn the generation of complex plans as well as plans aligned with human preferences.

|  | AV-A | AV-T | AV-V | IA-T | IA-V | IV-A | IV-T | IV-V | MA-I |
|---|---|---|---|---|---|---|---|---|---|
| **GPT4o** | 100 | 100 | 100 | 100 | 100 | 100 | 100 | 100 | 100 |
| **MultiMedia-Agent-1** | 80 | 50 | 50 | 70 | 50 | 60 | 70 | 80 | 90 |
| **MultiMedia-Agent-2** | 100 | 90 | 80 | 100 | 90 | 90 | 90 | 100 | 100 |
| **MultiMedia-Agent-3** | 100 | 90 | 40 | 100 | 60 | 80 | 100 | 40 | 90 |

|  | MA-T | MA-V | MI-A | MI-T | MI-V | MV-A | MV-I | MV-T | MV-V |
|---|---|---|---|---|---|---|---|---|---|
| **GPT4o** | 100 | 60 | 100 | 100 | 70 | 100 | 90 | 100 | 80 |
| **MultiMedia-Agent-1** | 90 | 10 | 70 | 80 | 50 | 50 | 70 | 60 | 60 |
| **MultiMedia-Agent-2** | 100 | 40 | 80 | 100 | 70 | 80 | 90 | 90 | 80 |
| **MultiMedia-Agent-3** | 70 | 30 | 50 | 90 | 40 | 90 | 80 | 90 | 80 |

Table 5: Comparison of success rate for GPT4o and MultiMedia-Agent.

|  | MA-T | MV-T | IV-T | MA-T | MI-T | MV-T |
|---|---|---|---|---|---|---|
| **GPT4o** | 4.2 | 4.2 | 4.5 | 4.2 | 4.8 | 4.8 |
| **MultiMedia-Agent-1** | 4.1 | 4.2 | 4.4 | 4.2 | 4.8 | 4.9 |
| **MultiMedia-Agent-2** | 4.1 | 4.2 | 4.4 | 4.0 | 4.8 | 4.8 |
| **MultiMedia-Agent-3** | 4.2 | 4.3 | 4.5 | 4.5 | 4.8 | 4.9 |

Table 6: Comparisons for text generation tasks with text output metrics.

### 4.3.2 VISUALIZATION RESULTS

As seen from the Figure 3, the plan generated by MultiMedia-Agent-1 lacks corresponding audio and special effects. MultiMedia-Agent-2 added sound effects to the plan, although they did not match the atmosphere of the video. In contrast, MultiMedia-Agent-3 generated content that included both subtitles and special effects, as well as appropriate ocean wave audio.

## 5 DISCUSSION

### 5.1 LIMITATION

Firstly, for tool selection, we currently use a prompt-based approach. However, considering the vast number of tools available in real-world scenarios, techniques like Retrieval-Augmented Generation (RAG) can be employed to optimize tool selection. Secondly, when it comes to solving complex tasks, multi-agent systems

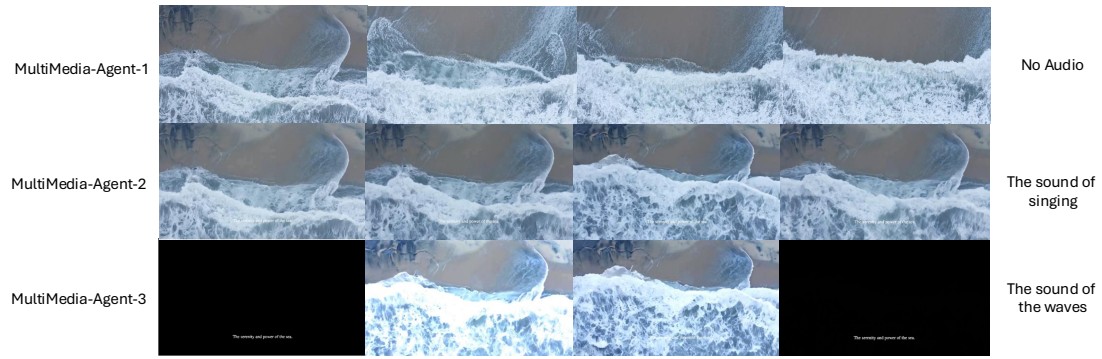

Figure 3: Visualization of the multimedia content created from the plan generated by MultiMedia-Agent. The user query is: use the images and the corresponding video to create a satisfying video.

| | MA-T | MV-T | IV-T | MA-T | MI-T | MV-T |
|---|---|---|---|---|---|---|
| **GPT4o** | 4.5 | 3.8 | 4.2 | 3.7 | 4.0 | 4.7 |
| **MultiMedia-Agent-1** | 4.4 | 3.7 | 4.0 | 3.8 | 4.1 | 4.5 |
| **MultiMedia-Agent-2** | 4.4 | 3.7 | 4.0 | 3.8 | 4.1 | 4.5 |
| **MultiMedia-Agent-3** | 4.6 | 3.9 | 4.1 | 3.9 | 4.2 | 4.6 |
| | **MA-T** | **MV-T** | **IV-T** | **MA-T** | **MI-T** | **MV-T** |
| **GPT4o** | 3.8 | 3.9 | 3.6 | 4.0 | 4.2 | 3.6 |
| **MultiMedia-Agent-1** | 3.8 | 3.7 | 3.6 | 4.1 | 4.1 | 3.8 |
| **MultiMedia-Agent-2** | 3.7 | 3.6 | 3.6 | 4.1 | 4.2 | 3.6 |
| **MultiMedia-Agent-3** | 4.3 | 3.9 | 3.8 | 4.3 | 4.1 | 3.9 |
| | **MA-T** | **MV-T** | **IV-T** | **MA-T** | **MI-T** | **MV-T** |
| **GPT4o** | 2.1 | 1.6 | 1.7 | 1.4 | 1.7 | 2.3 |
| **MultiMedia-Agent-1** | 2.0 | 1.6 | 1.8 | 1.3 | 1.6 | 2.2 |
| **MultiMedia-Agent-2** | 2.0 | 1.5 | 1.8 | 1.5 | 1.4 | 2.2 |
| **MultiMedia-Agent-3** | 2.0 | 1.9 | 1.8 | 1.6 | 1.8 | 2.2 |
| | **MA-T** | **MV-T** | **IV-T** | **MA-T** | **MI-T** | **MV-T** |
| **GPT4o** | 3.6 | 3.8 | 3.1 | 3.9 | 3.1 | 3.2 |
| **MultiMedia-Agent-1** | 3.3 | 3.9 | 3.2 | 3.9 | 3.2 | 3.2 |
| **MultiMedia-Agent-2** | 3.3 | 3.9 | 3.0 | 4.0 | 3.2 | 3.1 |
| **MultiMedia-Agent-3** | 3.9 | 3.9 | 3.6 | 4.2 | 3.3 | 3.5 |
| | **MA-T** | **MV-T** | **IV-T** | **MA-T** | **MI-T** | **MV-T** |
| **GPT4o** | 2.8 | 2.9 | 2.9 | 3.2 | 3.0 | 3.0 |
| **MultiMedia-Agent-1** | 3.0 | 2.9 | 2.7 | 3.4 | 3.0 | 3.1 |
| **MultiMedia-Agent-2** | 2.9 | 2.8 | 2.7 | 3.4 | 3.0 | 3.1 |
| **MultiMedia-Agent-3** | 3.1 | 3.1 | 2.9 | 3.4 | 3.2 | 3.2 |
| | **MA-T** | **MV-T** | **IV-T** | **MA-T** | **MI-T** | **MV-T** |
| **GPT4o** | 4.1 | 4.2 | 3.5 | 4.7 | 3.8 | 3.9 |
| **MultiMedia-Agent-1** | 3.9 | 4.1 | 3.4 | 4.6 | 3.9 | 4.0 |
| **MultiMedia-Agent-2** | 3.9 | 4.2 | 3.3 | 4.5 | 3.9 | 3.9 |
| **MultiMedia-Agent-3** | 4.2 | 4.1 | 3.6 | 4.8 | 3.9 | 4.1 |

Table 7: Comparisons for video generation tasks with video output metrics. From top to down: *Video Human Alignment; Video Psychological Appealing; Video Aestheic Score; Audio Human Alignment; Audio Psychological Appealing; Audio Video Alignment.*

are generally more effective than single-agent systems. In our future work, we plan to explore the use of multi-agent systems to tackle complex content generation tasks.

## 5.2 CONCLUSION

In this paper, we design a multimedia content generation agent system that leverages the skill acquisition theory to significantly enhance the capabilities of AIGC technologies in creating complex, multimodal content. By integrating a robust data pipeline, diverse tool library, and innovative evaluation metrics, our approach not only refines the content generation process but also aligns it more closely with real-world applications. The deployment of our MultiMedia-Agent, which outperforms traditional models like GPT4o, showcases the effectiveness of embedding skill acquisition into AI training regimens. This paves the way for further advancements in automated content creation, promising richer and more effective multimedia outputs.

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

# A APPENDIX

## A.1 TOOL LIBRARY

We present all the tools we used for the agent system in Table 8 . Including tools for image, video, audio, speech and text, Also includes task type such as editing, generation, retrieval, etc.

| Tool Name | Corresponding Model(s) |
|---|---|
| speech_mp3_to_text_txt | openai_wisper |
| audio_mp3_to_text_txt | audio_to_text |
| image_png_to_text_txt | openai_gpt4o, image_to_text |
| video_mp4_to_text_txt | openai_gpt4o, video_to_text |
| text_txt_to_txt_txt | openai_gpt4o, text_to_text |
| text_txt_to_speech_mp3 | text_to_speech |
| text_txt_to_image_png | text_to_image |
| text_txt_to_audio_mp3 | text_to_audio |
| text_txt_and_image_png_to_video_mp4 | text_and_image_to_video |
| image_png_quality_assessment | openai_gpt4o, image_quality_assessment |
| video_mp4_quality_assessment | openai_gpt4o, video_quality_assessment |
| audio_mp3_text_alignment | audio_to_text, openai_gpt4o |
| retrieve_image_from_web_to_image_png | search_images |
| retrieve_video_from_web_to_video_mp4 | search_videos |
| retrieve_audio_from_web_to_audio_mp3 | retrieve_audio_from_web_to_audio_mp3 |
| text_txt_image_png_object_removal_to_image_png | instruct_pix2pix, Remove the |
| image_png_watermark_removal_to_image_png | instruct_pix2pix, Remove the watermark |
| text_txt_image_png_object_adding_to_image_png | instruct_pix2pix, Add |
| video_mp4_object_removal_to_video_mp4 | instruct_pix2pix, Remove the |
| video_mp4_watermark_removal_to_video_mp4 | instruct_pix2pix, Remove the watermark |
| video_mp4_object_adding_to_video_mp4 | instruct_pix2pix, Add |
| image_png_crop_base_on_text_to_image_png | image_crop_base_on_text |
| image_png_desnowing_to_image_png | diff_plugin_img, desnow |
| image_png_dehazing_to_image_png | diff_plugin_img, dehaze |
| image_png_deblurring_to_image_png | diff_plugin_img, deblur |
| image_png_deraining_to_image_png | diff_plugin_img, derain |
| image_png_face_restoration_to_image_png | diff_plugin_img, face |
| image_png_demoreing_to_image_png | diff_plugin_img, demoire |
| image_png_low_light_enhancement_to_image_png | image_low_light_enhance |
| video_mp4_desnowing_to_video_mp4 | diff_plugin_vid, desnow |
| video_mp4_dehazing_to_video_mp4 | diff_plugin_vid, dehaze |
| video_mp4_deblurring_to_video_mp4 | diff_plugin_vid, deblur |
| video_mp4_deraining_to_video_mp4 | diff_plugin_vid, derain |
| video_mp4_face_restoration_to_video_mp4 | diff_plugin_vid, face |
| video_mp4_demoreing_to_video_mp4 | diff_plugin_vid, demoire |
| video_mp4_cut_to_video_mp4 | video_cut |
| video_mp4_key_frame_to_image_png | video_key_frame_to_image |
| video_mp4_extraction_to_audio_mp3 | video_extract_audio |
| video_mp4_super_resolution_to_video_mp4 | video_super_resolution_to_video |
| image_png_super_resolution_to_image_png | image_super_resolution_to_image |

| image_png_video_concatenate_to_video_mp4 | image_video_concatenate |
|---|---|
| video_mp4_video_concatenate_to_video_mp4 | video_video_concatenate |
| video_mp4_audio_concatenate_to_video_mp4 | video_audio_concatenate |
| video_mp4_subtitle_concatenate_to_video_mp4 | video_sutitle_concatenate |
| video_mp4_speed_up_to_video_mp4 | clip.fx(vfx.speedx, factor) |
| video_mp4_speed_down_to_video_mp4 | clip.fx(vfx.speedx, 1/factor) |
| effect_video_mp4_fade_to_video_mp4 | moviepy_video, fade |
| effect_video_mp4_horizontal_mirror_to_video_mp4 | moviepy_video, horizontal_mirror |
| effect_video_mp4_vertical_mirror_to_video_mp4 | moviepy_video, vertical_mirror |
| effect_video_mp4_brightness_adjustment_to_video_mp4 | moviepy_video, brightness_adjustment |
| effect_video_mp4_change_black_and_white_to_video_mp4 | moviepy_video, black_and_white |
| audio_mp3_audio_concatenate_to_audio_mp3 | sox_audio, audio_concatenate |
| audio_mp3_speed_up_to_audio_mp3 | sox_audio, speed_up |
| audio_mp3_speed_down_to_audio_mp3 | sox_audio, speed_down |
| audio_mp3_change_volume_to_audio_mp3 | sox_audio, change_volume |
| effect_audio_mp3_add_reverb_to_audio_mp3 | sox_audio, add_reverb |
| effect_audio_mp3_add_echo_to_audio_mp3 | sox_audio, add_echo |
| effect_audio_mp3_fade_in_to_audio_mp3 | sox_audio, fade_in |
| effect_audio_mp3_fade_out_to_audio_mp3 | sox_audio, fade_out |
| effect_audio_mp3_add_stereo_widening_to_audio_mp3 | sox_audio, add_stereo_widening |
| text_txt_image_png_object_detection_to_image_png | image_object_detection |
| image_png_resize_to_image_png | image_ffmpeg, image_resize |
| effect_image_png_rotate_to_image_png | image_ffmpeg, image_rotate |
| effect_image_png_vertical_flip_to_image_png | image_ffmpeg, image_vertical_flip |
| effect_image_png_horizontal_flip_to_image_png | image_ffmpeg, image_horizontal_flip |
| effect_image_png_trim_to_image_png | image_ffmpeg, image_trim |
| effect_image_png_change_brightness_to_image_png | image_ffmpeg, image_change_brightness |
| effect_image_png_contrast_to_image_png | image_ffmpeg, image_contrast |
| effect_image_png_saturation_to_image_png | image_ffmpeg, image_saturation |
| effect_image_png_hue_to_image_png | image_ffmpeg, image_hue |
| effect_image_png_blur_to_image_png | image_ffmpeg, image_blur |
| effect_image_png_sharpen_to_image_png | image_ffmpeg, image_sharpen |
| effect_image_png_noise_to_image_png | image_ffmpeg, image_noise |
| effect_image_png_emboss_to_image_png | image_ffmpeg, image_emboss |
| effect_image_png_sketch_to_image_png | image_ffmpeg, image_sketch |
| effect_image_png_mosaic_to_image_png | image_ffmpeg, image_mosaic |
| image_png_add_subtitle_to_image_png | add_caption_to_image_png |
| add_subtitle_to_video_mp4 | add_subtitle_to_video_mp4 |

Table 8: Tool Library

## A.2 Example of generated plan

We first show the format of the tool library; tool calling function and Generated plan in the Figure. 4. Then we show an example plan in the Table. A.2.

- **Step 1: Text to Speech**
    - **Tool**: text_txt_to_speech_mp3

- **Instruction**: The ocean at sunset is a tranquil and mesmerizing scene, evoking a sense of calm and reflection.
- **Output**: audio_1_0.mp3

• **Step 2: Create Video from Image and Text**
  - **Tool**: text_txt_and_image_png_to_video_mp4
  - **Input**: mixkit-sea-waves-reflecting-the-sunset-1927_1.png
  - **Output**: video_2_0.mp4

• **Step 3: Create Video from Another Image and Text**
  - **Tool**: text_txt_and_image_png_to_video_mp4
  - **Input**: mixkit-sea-waves-reflecting-the-sunset-1927_2.png
  - **Output**: video_3_0.mp4

• **Step 4: Create Video from a Third Image and Text**
  - **Tool**: text_txt_and_image_png_to_video_mp4
  - **Input**: mixkit-sea-waves-reflecting-the-sunset-1927_3.png
  - **Output**: video_4_0.mp4

• **Step 5: Concatenate Videos (Step 2 and Step 3)**
  - **Tool**: video_mp4_video_concatenate_to_video_mp4
  - **Input**: video_2_0.mp4, video_3_0.mp4
  - **Output**: video_5_0.mp4
  - **Depends**: 2, 3

• **Step 6: Concatenate with Another Video (Step 5 and Step 4)**
  - **Tool**: video_mp4_video_concatenate_to_video_mp4
  - **Input**: video_5_0.mp4, video_4_0.mp4
  - **Output**: video_6_0.mp4
  - **Depends**: 5, 4

• **Step 7: Apply Fade Effect**
  - **Tool**: effect_video_mp4_fade_to_video_mp4
  - **Input**: video_6_0.mp4
  - **Output**: video_7_0.mp4
  - **Depends**: 6

• **Step 8: Add Audio to Video**
  - **Tool**: video_mp4_audio_concatenate_to_video_mp4
  - **Input**: video_7_0.mp4, audio_1_0.mp3
  - **Output**: video_8_0.mp4
  - **Depends**: 7, 1

• **Step 9: Add Subtitle**
  - **Tool**: video_mp4_subtitle_concatenate_to_video_mp4
  - **Instruction**: The ocean at sunset is a tranquil and mesmerizing scene, evoking a sense of calm and reflection.
  - **Input**: video_8_0.mp4
  - **Output**: video_9_0.mp4
  - **Depends**: 8

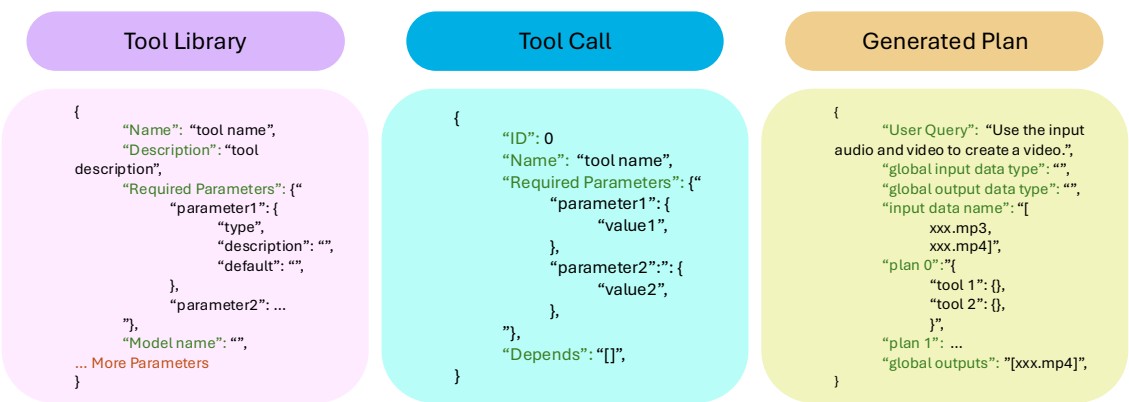

Figure 4: Formats for tool library and generated plan.

## A.3 TRAINING DETAILS OF MULTIMEDIA-AGENT

For the first stage, we train the Minicpm-v2 with a learning rate of $2e - 6$. Weight decay is $0.1$, training step is $10000$, warmup ratio is $0.01$. For the second stage, we degrade the training step into $2000$ and the learning rate to $1e - 6$. For the third stage, we degrade the training step to $1000$ and the learning rate to $5e - 7$, all the experiments were conducted on $4 \times A5000$ GPU.

## A.4 RESULTS FOR IMAGE AND AUDIO GENERATION

Here, we present the results for image generation and audio generation. For the image generation results, the metrics from top to bottom are:*Image Human Alignment, Image Psychological Appeal, and Image Aesthetic Score*. For the audio generation results, the metrics from top to bottom are: *Audio Human Alignment and Audio Psychological Appeal*.

|  | MA-I | MV-I |
|---|---|---|
| **GPT4o** | 4.0 | 4.5 |
| **MultiMedia-Agent-1** | 4.1 | 4.2 |
| **MultiMedia-Agent-2** | 4.2 | 4.1 |
| **MultiMedia-Agent-3** | 4.5 | 4.6 |
|  | MA-I | MV-I |
| **GPT4o** | 3.8 | 4.0 |
| **MultiMedia-Agent-1** | 3.8 | 3.5 |
| **MultiMedia-Agent-2** | 3.7 | 3.4 |
| **MultiMedia-Agent-3** | 4.0 | 4.0 |
|  | MA-I | MV-I |
| **GPT4o** | 6.2 | 7.1 |
| **MultiMedia-Agent-1** | 6.3 | 7.4 |
| **MultiMedia-Agent-2** | 6.1 | 7.3 |
| **MultiMedia-Agent-3** | 6.5 | 7.5 |

|  | AV-A | IV-A | MI-A | MV-A |
|---|---|---|---|---|
| **GPT4o** | 4.3 | 4.0 | 3.5 | 3.5 |
| **MultiMedia-Agent-1** | 4.2 | 3.8 | 3.6 | 3.6 |
| **MultiMedia-Agent-2** | 4.3 | 3.7 | 3.4 | 3.5 |
| **MultiMedia-Agent-3** | 4.5 | 3.9 | 3.5 | 3.6 |

|  | AV-A | IV-A | MI-A | MV-A |
|---|---|---|---|---|
| **GPT4o** | 4.3 | 3.8 | 3.5 | 3.4 |
| **MultiMedia-Agent-1** | 4.0 | 3.7 | 3.6 | 3.5 |
| **MultiMedia-Agent-2** | 3.7 | 3.6 | 3.5 | 3.5 |
| **MultiMedia-Agent-3** | 4.4 | 3.6 | 3.7 | 3.7 |

