# OpenReview forum: "MultiMedia-Agent: A Multimodal Agent for Multimedia Content Generation"
_ICLR.cc/2025/Conference — ICLR 2025 Conference Withdrawn Submission_

### Official Review · Reviewer_1igN · 2024-10-25

**Soundness:** 1
**Presentation:** 1
**Contribution:** 2
**Rating:** 3
**Confidence:** 4

**Summary:**

This paper introduces a multimedia agent for content generation agent. It first proposes a data generation pipeline, a tool library and several evaluation metrics. A two-stage correlation plan curation method and a three-stage training pipeline are proposed according to the skill acquisition theory. The authors conduct experiments to compare its performance against GPT-4o.

**Strengths:**

1. The general idea of multimodal generation agent with tool using and planning is interesting.
2. The proposed method covers a wide range of tasks and tools.

**Weaknesses:**

This paper is poorly written. The experiments are also not convincing.

1. The only compared baseline is GPT-4o, which is not specifically designed for most of the tasks. More baselines should be added, such as those in Table 1, even if they are not applicable for all of the tasks. It is also not clear how the GPT-4o baseline is implemented for other tasks like audio generation or video generation.
2. Samples of generation are not sufficient. The only provided demo is Figure 9, where the audio is indirectly presented as text descriptions. Demos for other tasks are not given.
3. The experiment improvements are trivial. Besides, there are only 10 queries for each of the tasks in the validation set. What are the confidence intervals? Are the results statistically significant?
4. The explanations of "longer plans" and "fewer steps" should not be concluded directly, but supported by additional experiments showing the average length of steps of each model.
5. What is the metric in Table 6? What are the meanings of the metrics in Table 7 and the two tables in the appendix? Key explanations are missing. Also, why are the tasks in Table 7 all text generation tasks? Shouldn't they be "xx-V"?
6. What are the details of the tools? Table 8 is not sufficient as entries like "audio_to_text", "text_to_image" are not detailed enough. For instance, what underlying models or algorithms are used? What are the input/output specifications and any key parameters?
7. What are the details of the metrics in Section 3.3.1? The current description is not enough for reproducibility.
8. Many typos and grammar mistakes throughout the paper.

**Questions:**

1. Why not use ImageBind for task formulation or evaluation?
2. What is the meaning of success rate? Does a failed plan mean it is not executable due to incorrect parameters, does not use the correct tools, or anything else?

---

### Official Review · Reviewer_pJVa · 2024-10-29

**Soundness:** 3
**Presentation:** 3
**Contribution:** 2
**Rating:** 5
**Confidence:** 3

**Summary:**

This paper introduces a multimedia content generation agent, referred to as the MultiMedia-Agent, which is designed to automate complex multimedia content creation tasks. The authors position their agent as a system capable of outperforming existing generative AI models in this space, including GPT-4o. Through comparative analysis, they argue that their proposed MultiMedia-Agent generates higher-quality multimedia content, offering a better media outputs compared to GPT-4o.

**Strengths:**

1.	The paper is well-structured and easy to follow, making its technical concepts accessible to readers, which enhances understanding and supports the proposed research’s coherence.

2.	The topic of multimedia content automation is timely and has high relevance, especially given the expanding demand for digital content across various domains, from marketing to education. This research holds considerable potential for real-world application, promising efficiency and scalability in daily content creation tasks.

3.	The authors’ attempt to specialize in multimedia content generation represents an innovative approach that could fill an important gap in automated content creation, potentially providing richer, multi-modal outputs beyond current text-based LLM capabilities.

**Weaknesses:**

1.	The framework appears to primarily leverage existing technologies without significant structural innovation. It’s unclear if the advancements lie in model architecture or simply application. Expanding on how the MultiMedia-Agent advances beyond the foundational technologies would strengthen the paper.

2.	The comparison with GPT-4o raises concerns, as GPT-4o is not explicitly designed for multimedia content generation. This choice limits the comparative relevance, as the study might benefit from benchmarking against more specialized or similar frameworks in multimedia generation. Adding such comparisons would enhance the credibility of the proposed system's advantages.

3.	I am a bit concerned about the evaluation metrics the authors proposed. It seems to be that most of the metrics are based on GPT-4o. It will be more convincing if the authors can show the evaluation from GPT-4o truly aligns with human perceptions.

4.	Minor typographical errors appear in the text, including the abstract. For instance, in the abstract, “the our approaches” should be revised to “our approaches” to maintain professionalism and clarity.

Minor Suggestions:
•	Including citations for comparison methods in Table 1 would allow readers to trace back the origins and contexts of these models, lending credibility and clarity.
•	Ensure consistent use of terms, such as “GPT4o” or “GPT-4o,” for a more polished presentation.

**Questions:**

1.	Large Language Models (LLMs) can exhibit unpredictable behavior, so showing examples of failure cases for the MultiMedia-Agent would add depth and transparency. Analyzing these cases could provide insight into potential improvements.

2.	Has the success rate of the MultiMedia-Agent been quantified? Understanding the model’s reliability across different types of content generation would strengthen the case for its practical application and offer a valuable metric for future benchmarking. Did the authors notice any bias issues during content generation?

---

### Official Review · Reviewer_Jrej · 2024-10-30

**Soundness:** 2
**Presentation:** 3
**Contribution:** 2
**Rating:** 3
**Confidence:** 4

**Summary:**

This paper introduced a multi-agent large language model (LLM) framework based on the Skill Acquisition Theory that supports any-to-any styled generation, including text, image, audio, and image. Such framework was evaluated based on model-based preference evaluation metrics. As the result of evaluation, the framework's best version (i.e. with 3 stages included) was able to show comparative performance with GPT-4o while the overall success rate is lower. In summary, this study was able to propose a relatively good multi-agent LLM framework with multiple components, which showed comparative performance as GPT-4o in certain aspects based on the metrics that the paper claimed.

**Strengths:**

1. The originality of this paper is worth noting. The idea of applying the Skill Acquisition Theory into the design of the framework is inspiring. Using information theory as a design guide when implementing multi-agent is a good thought other than just adding multiple iterations naively. I personally feel this is a really interesting idea and definitely would like to see more related work in future.
2. The paper's structure is very clear and easy to follow. The quality of the overall presentation is pretty good.

**Weaknesses:**

Unfortunately, I will have to vote for reject for this paper as it has some fundamental flaws in its evaluation.

1. The evaluation metrics is not a solid one. Although the idea of this paper might looks theoretically beautiful, its experiments lack convincing support. For content generation, especially when LLM is involved, there has been tremendous excellent studies where multiple kinds of evaluation have been introduced. For example, when handling artistic or abstract content generation (e.g. music/audio/image), it would be hard to solely rely on LLM model evaluation as LLM model could have certain problems such as LLM hallucination and unfortunately these problems are still pending on being solved/studied. Therefore, subjective evaluation is currently still necessary to evaluate generated content especially for **content matching task**, such as AB test, ranking test, or rating test. This could easily and intuitively show the advantage of each model based on a large group of experts/users' feeling/rating. Several studies on recent top conferences have made remarkable examples on such kind of evaluation such as [1][2][3].
2. The comparative study is not solid enough. This paper only compare the framework with GPT-4o. To make this paper in a better shape for publishment, it will need to include more relevant model/framework for comparison.

[1] Yue, Xiang, et al. "Mmmu: A massive multi-discipline multimodal understanding and reasoning benchmark for expert agi." Proceedings of the IEEE/CVF Conference on Computer Vision and Pattern Recognition. 2024.
[2] Deng, Qixin, et al. "ComposerX: Multi-Agent Symbolic Music Composition with LLMs." arXiv preprint arXiv:2404.18081 (2024).
[3] Guo, T., et al. "Large Language Model based Multi-Agents: A Survey of Progress and Challenges." 33rd International Joint Conference on Artificial Intelligence (IJCAI 2024). IJCAI; Cornell arxiv, 2024.

**Questions:**

As mentioned as above, although I like the idea of this paper and enjoy reading it, I will have to reject it. To make this paper in a better shape for publihsment, I would recommend as below.

1. Have a more thorough study on relevant studies and try to include them in the comparison/evaluation section.
2. Improve evaluation metrics and include more convincing experimental results on the superiority of the framework.

---

### Official Review · Reviewer_gKyB · 2024-11-04

**Soundness:** 2
**Presentation:** 2
**Contribution:** 2
**Rating:** 5
**Confidence:** 4

**Summary:**

I am very motivated by this article because it indeed addresses a real issue. However, as with other studies in this research path, the validation of the experiments is very weak. I hope the authors can discuss this in the discussion period.

**Strengths:**

I believe the advantages of this type of article are self-evident and are enough to impact the industry. Therefore, compared to the advantages, I hope to discuss more about the missing parts.

**Weaknesses:**

- This article seems more like a prototype design rather than a complete paper, as it lacks many implementation and experimental details.
- I didn't see any examples, nor did I see any supplementary materials provided for demonstration (did I miss something?).
- How is the success rate validated? How is success defined?
- I understand A stands for audio, and V stands for video, but what does AV-V mean? What is the task? What is the goal? Does it require - - human involvement, as the paper mentions human alignment as a contribution?
- What are Plan1, Plan2, and Plan3? What are the differences?
- What do Agent1, Agent2, and Agent3 represent? What is their significance?
- What does Average steps mean? Is fewer better?
- What are the differences in the success rate between Tables 4 and 5?
- Each task seems to have different input/output formats. How are they validated separately?
- The images look very rudimentary, and some of the text is even unclear.

**Questions:**

Refer to weaknesses.

---

### Author Response · Authors · 2024-11-19
**Response to all reviewers**

Thank you to all the reviewers for your appreciation and valuable suggestions. We have carefully studied each of your comments. We will design a more standardized and fair evaluation scheme and revise the paper thoroughly based on your feedback. We hope that the revised version will better reflect the value of our research and meet your expectations. Once again, thank you for your support and assistance with our work!

---

### Note · Authors · 2024-11-19

I have read and agree with the venue's withdrawal policy on behalf of myself and my co-authors.